# From Structural to Functional Hypertension Mediated Target Organ Damage—A Long Way to Heart Failure with Preserved Ejection Fraction

**DOI:** 10.3390/jcm11185377

**Published:** 2022-09-13

**Authors:** Costantino Mancusi, Maria Lembo, Maria Virginia Manzi, Christian Basile, Ilaria Fucile, Carmine Morisco

**Affiliations:** Department of Advanced Biomedical Sciences, Hypertension Research Center, University Federico II of Naples, Via Sergio Pansini 5, 84031 Naples, Italy

**Keywords:** arterial hypertension, heart failure, left ventricular hypertrophy, coronary microvascular dysfunction, systolic dysfunction, microalbuminuria

## Abstract

Arterial hypertension (AH) is a major risk factor for the development of heart failure (HF) which represents one of the leading causes of mortality and morbidity worldwide. The chronic hemodynamic overload induced by AH is responsible for different types of functional and morphological adaptation of the cardiovascular system, defined as hypertensive mediated target organ damage (HMOD), whose identification is of fundamental importance for diagnostic and prognostic purposes. Among HMODs, left ventricular hypertrophy (LVH), coronary microvascular dysfunction (CMVD), and subclinical systolic dysfunction have been shown to play a role in the pathogenesis of HF and represent promising therapeutic targets. Furthermore, LVH represents a strong predictor of cardiovascular events in hypertensive patients, influencing per se the development of CMVD and systolic dysfunction. Clinical evidence suggests considering LVH as a diagnostic marker for HF with preserved ejection fraction (HFpEF). Several studies have also shown that microalbuminuria, a parameter of abnormal renal function, is implicated in the development of HFpEF and in predicting the prognosis of patients with HF. The present review highlights recent evidence on the main HMOD, focusing in particular on LVH, CMD, subclinical systolic dysfunction, and microalbuminuria leading to HFpEF.

## 1. Introduction

Identification of hypertensive mediated target organ damage (HMOD) is of paramount importance for diagnostic and prognostic purposes [1]. During the last ten years, evidence supports the hypothesis that the development of HMOD is a multifactorial process which involves both demographic and clinical characteristics of hypertensive patients. Presence of left ventricular hypertrophy (LVH) is currently defined as a structural HMOD which per se might influence the development of functional HMOD represented by coronary microvascular dysfunction and subclinical systolic dysfunction. The presence of the above-mentioned conditions is particularly evident among patients affected by heart failure with preserved ejection fraction (HFpEF). The present review highlights recent evidence on the main HMOD, focusing in particular on LVH, coronary microvascular dysfunction, renal impairment, and subclinical systolic dysfunction leading to HFpEF.

## 2. Left Ventricular Hypertrophy

Onset and perseverance of LVH is a strong marker for adverse cardiovascular prognosis in hypertensive patients [2,3]. In the presence of increased afterload due to elevated blood pressure (BP), according to Laplace’s Law [LV wall stress = (LV pressure × LV radius)/(2 × LV wall thickness)], an increase in LV wall thickness lowers the tension acting on the individual myocardial cell. Prevalence of LVH ranges from 20% in mildly hypertensive patients to almost 100% in those with severe or complicated arterial hypertension [4]. Different epidemiological studies have demonstrated that BP components have a direct impact on the development of LVH, highlighting that BP control has a crucial role in its prevention [5,6]. Pulse pressure has a strong impact in determining the development of LVH, but data from epidemiological studies suggest that systolic BP is the link in the relation between pulse pressure and LVH [6]. Different population-based studies have highlighted that ambulatory blood pressure measurements (ABPM) are more closely related to LVH than conventional BP readings taken in the clinician’s office [7]. Furthermore, recent evidence from the Campania Salute Network registry identified BP variability as one of the main contributors to development of LVH [8,9].

To assess the presence of LVH, LV mass needs to be normalized for body size. This was conventionally accomplished using body surface area (BSA), but, on the basis of large quantities of evidence [10,11], it is discouraged today, albeit still largely used, because this method underestimates LV mass in obese subjects [12]. Normalization for height in meters raised to the power of 2.7, called allometric signal, proved to be more appropriate [13]. This method maximizes population risk attributable to LVH when prevalence of overweight/obesity is high, a common finding in middle-aged patients with arterial hypertension. BSA may still be used in normal-weight patients [14].

Other than the impact of BP, other demographical and epidemiological aspects are crucial for the development and maintenance of LVH. We have previously demonstrated that LVH progression/regression is strongly related to the female sex and the presence of obesity [15,16]. Consistently, in individuals with a new diagnosis of hypertension, obesity represents the most relevant risk factor for LVH [17]. Losing weight is one of the crucial aspects, beyond adequate BP control, which may help the regression of LVH [16,18,19,20].

LVH is the most common myocardial structural abnormality associated with HFpEF, and clinical evidence links LVH to diastolic dysfunction and qualifies LVH as a diagnostic marker for HFpEF [21], albeit not being present in the H2FPEF score [22]. The more recent guideline of HF by ESC highlights the importance of LV mass as a main morphological criterion for the diagnosis of HFpEF [23].

Previous studies have suggested that patients with LVH and preserved ejection fraction may have subtle systolic dysfunction not reflected in the ejection fraction evaluation [24]. Altered contractility in LVH is related to structural and functional abnormalities involving the extracellular matrix, fibrous tissue, and the vasculature as well as the cardiomyocytes themselves. Different animal studies have also highlighted that microvascular dysfunction and vascular remodeling associated with LVH may promote HFpEF [25,26].

LVH contributes to the remodeling of the left atrium, acting as a trigger for the development and maintenance of atrial fibrillation (AF). Patients with concentric LVH have more significant diastolic dysfunction leading to chronic elevation of filling pressures, which in turns results in left atrial dilatation and development of AF [27,28,29]. Furthermore, different patterns of LVH have been demonstrated to be important determinants of cardiovascular outcomes in patients with AF, contributing to the development of adverse cardiovascular outcomes including overt HFpEF [30]. The impact of LVH on diastolic dysfunction has been demonstrated by several studies focusing on the general population as well as hypertensive patients and is associated with worsened cardiovascular prognosis [31,32]. Myocardial hypertrophy leads to an increase of the myocardial mass/volume ratio, and the degree of hypertrophy is the main determinant of chamber stiffness. The main, if not unique, determinant of myocardial diastolic tissue distensibility is the structure and concentration of the collagen. LV collagen concentration is elevated due to reactive fibrosis. Studies using MRI have demonstrated that development of fibrosis precedes the overt HF and might be considered as an important contributor to its development [33].

## 3. Coronary Microvascular Dysfunction

Hypertension is one of the major causes of ischemic heart disease (IHD) and HFpEF [1]. The pathogenic mechanism that accounts for the IHD is the development of atherosclerosis which is, in fact, considered a manifestation of HMOD. The recent progresses in invasive and non-invasive cardiovascular imaging allowed additional mechanisms that play a role in the pathogenesis of IHD and HFpEF to be defined. The coronary microvascular dysfunction (CMVD) is defined by concomitant presence of the following criteria: (a) symptoms or documentation of IHD, (b) evidence of non-significant epicardial coronary artery disease, and (c) evidence of reduced coronary blood flow [34].

CMVD increases cardiovascular morbidity and mortality [35], negatively affects the prognosis of IHD even in presence of successfully percutaneous intervention [36,37], plays a role in the pathogenesis of HFpEF [26,38], and represents a promising target of therapies in myocardial infarction [39,40]. The clinical manifestations of CMVD mainly consist of IHD without obstruction of coronary arteries, acute myocardial infarction without obstruction of coronary arteries, takotsubo syndrome, and HFpEF [41].

Female gender, age, hypertension, smoking status, enhanced resting heart rate, low levels of high-density-lipoproteins (HDL) cholesterol, and diabetes are the principal determinants of CMVD [41,42].

The microvascular dysfunction in hypertension is a well-documented phenomenon; for instance, it is responsible of hypertensive retinopathy [43] and contributes to the development of chronic kidney disease (CKD) [44] and cognitive impairment [45]. In addition, a decreased coronary flow reserve (CFR) in the absence of epicardial coronary artery disease, a hallmark of CMVD, has also been reported in patients with hypertension or pre-hypertension [46,47]. The PROMIS-HFpEF trial demonstrated a high prevalence of coronary microvascular dysfunction in HFpEF in the absence of macrovascular coronary artery disease [48].

LVH, leading to the hypertrophy of vascular smooth muscle cells, interstitial and perivascular fibrosis, reduction of capillary density, endothelial dysfunction, and enhanced left ventricular filling pressure, plays a key role in the pathogenesis of CMVD [38].

The clinical consequences of CMVD highlight the need for a new approach in the stratification of CV risk in hypertensive patients. In particular, the CMVD should be considered a further manifestation of HMOD, and, thus, patients with hypertension at high risk for CV events should be screened for CMVD. Although the gold standard for the identification of CMVD includes measurements based on coronary angiography, even non-invasive methods allow for the assessment of myocardial blood flow (MBF). In particular, cardiac magnetic resonance (CMR) as well as cardiac PET allow for the measurement of both global and regional MBF. Although CMR avoids the exposure to the radiations, both techniques are time-consuming and expensive; therefore, they cannot be used in the ordinary clinical practice. The myocardial contrast echocardiography (MCE) evaluates the myocardial distribution of echographic contrast vehicles. The advantages of this techniques are the low costs and the large feasibility; unfortunately, to date, it was tested in only a few studies. Finally, the transthoracic Doppler echocardiography (TTDE) allows for the estimation of blood flow in the mid-part of the left anterior descending coronary artery by a pulsed Doppler wave; the advantages of this method are the feasibility and the reasonable costs. However, this technique still remains operator- and patient-dependent [49].

Improvement of CMVD represents a potential target of antihypertensive therapy. Several studies indicate that the inhibition of the renin-angiotensin-aldosterone system (RAAS) by either ACE-inhibitors or angiotensin II receptor antagonists exerts beneficial effects on HMDO. Unfortunately, there are only a few studies that specifically evaluated the effects of inhibition of RAAS on CMVD [50,51]. Recent studies have demonstrated the potential role of specific pharmacological therapy for microvascular dysfunction in HFpEF. In particular, data from animal studies has demonstrated that agents targeting NO-activated guanylyl cyclase (GC) might have a beneficial effect on microvascular dysfunction in HFpEF [52]. Recently, vericiguat, a once-daily stimulator of sGC, was tested in a population of patients with preserved systolic function. Unfortunately, the primary endpoint, defined as the change in NT-proBNP level, was negative, but there was evidence of improved physical capacity and quality of life [53]. Promising therapy with SGLT2 inhibitors has been demonstrated to improve microvascular dysfunction in animal models acting through the nitric oxide pathway [54]. This aspect needs to be deeply evaluated in further studies.

## 4. Systolic Dysfunction

Elevated blood pressure levels determine both altered myocardial relaxation and left ventricular (LV) systolic dysfunction by several mechanisms including increased afterload, myocardial ischemia, and myocardial fibrosis [55,56,57]. Even though LV diastolic dysfunction generally precedes LV systolic impairment [58,59], some precociously altered parameters derived from both standard and advanced echocardiography allow for the identification of an early systolic impairment.

Two-dimensional echocardiographic-derived LV ejection fraction (LVEF), the traditionally most commonly used parameter for LV systolic function evaluation, has the ability to have high variability and to be reduced when cardiac damage and dysfunction are already established such as in clinically overt stages of hypertensive heart disease [60,61] (Table 1).

Nonetheless, standard echocardiography provides additional parameters useful for determining an early systolic impairment in the hypertensive setting (Table 2). 

A reduction in stroke volume, an index of LV pump performance, was detected in hypertensive patients with LV concentric geometry [62]. Midwall fractional shortening has demonstrated the ability to identify early LV systolic dysfunction in hypertensive patients with normal LVEF [63], its alteration being more pronounced in patients with LV concentric hypertrophy [64]. Another parameter of LV systolic performance is represented by mechano-energetic efficiency (MEEi) which defines the magnitude of LV work developed for a given unit of energetic consumption; it is computed as the ratio between stroke work and oxygen consumption, indexed for LV mass [65,66]. MEEi was proved to be altered before LVEF reduction and was demonstrated to be a prognosticator of adverse cardiovascular events in the hypertensive setting [67,68]. Low MEEi was also demonstrated to be a determinant of future EF reduction in hypertensive patients [69]. Recent evidence from the Strong Heart Study has highlighted its utility as a prognostic marker for development of HF [44].

Furthermore, advanced ultrasound techniques are emerging for their capability in identifying subclinical LV systolic impairment, easy approach, and high accuracy [70]. In particular, speckle tracking echocardiography allows for the measurement of LV deformation in different directions: longitudinal, circumferential, and radial strains as well as LV twisting. All strains resulted in being altered in hypertensive heart disease [71]. Indeed, global longitudinal strain (GLS) represents a reproducible and sensitive parameter for detection of LV subclinical systolic dysfunction; despite being load dependent, it results in impairment even at very early stages of hypertension, before modifications in both LV geometry and LVEF [72]. Lower values of GLS in absolute value were also associated with the extension of myocardial fibrosis and to LV filling pressures [73]. GLS was demonstrated to be a predictor of major adverse cardiovascular events in a population of hypertensive patients when integrated in a risk score, including age greater than 70 years, LV concentric hypertrophy, and atrial fibrillation [74]. Moreover, in a population of newly-diagnosed hypertensive patients without LVH, regional longitudinal strain analysis described an exacerbation of the base-to-apex gradient with a relative sparing of apex, reflecting the segmental distribution of myocardial fibrosis in more advanced stages of hypertension in patients with concentric LVH (Figure 1) [75,76]. GLS progressively worsened in more compromised hypertensive patients presenting left ventricular concentric remodeling, LVH, and dilated hypertrophic ventricles [71].

Circumferential and radial strain deformations, on the other hand, resulted in being impaired in more advanced stages of hypertension and being more pronounced after development of LVH [77]. This phenomenon seems related to the progressive accumulation of pathologic fibrosis and disarray of the connective tissue matrix of interstitial collagen fibers in LVH, impacting LV circumferential contraction and disturbing radial thickening [78].

Indeed, LVH could impact LV function in multiple ways, with morphological aspects being reflected by functional alterations. LVH induced by arterial hypertension is accompanied by vascular and interstitial remodelling and fibrosis leading to progressive augmented microvascular resistance and arterial stiffness, endothelial disfunction, and impaired capillary density; it can also affect the coronary district [79]. All these abnormalities could contribute to triggering myocardial remodelling and apoptosis, myocardial ischemia, and progressively determine LV dilation and drop of LV systolic performance [80]. In fact, the natural history of hypertensive heart disease is related to the lost balance of what started as a compensatory mechanism to pressure overload and evolved towards progressive stages including abnormal LVH, diastolic impairment, HFpEF, LV dilation, and heart failure with reduced ejection fraction [81,82,83,84]. The study of myocardial deformation layers highlighted that all the layers of endo-, mid-, and subepicardial in both longitudinal and circumferential strains were altered in hypertensive patients, the longitudinal endomyocardial layer being the one more severely impaired [85,86]. In more advanced stages of pressure overload, subepicardial layer strain seemed to be more affected, the dysfunction of the latter also representing a prognosticator of cardiovascular events [87]. The interrelation and contemporary alteration of multiple strains associated with the alteration of echo parameters examining different myocardial layers—as demonstrated, for example, by the independent association between midwall fractional shortening (identifying the motion of midwall circumferential fibers) and GLS (evaluating the deformation of longitudinal fibers)—underlines the comprehensive LV systolic impairment in arterial hypertension, evident from very early stages [63,71].

In addition, three-dimensional echocardiography, besides the computation of LVEF and LV geometry, can provide the analysis of strain imaging with evaluation of the entire LV from a single volume of data acquisition. Even with this technique, an impairment in all strain deformations was detectable in hypertensive heart disease; this also included the alteration of global area strain, a strain peculiar to 3D assessment, representing the percentage change of LV wall from its original dimensions and, thus, a combination of circumferential and longitudinal deformations [88,89].

Therefore, arterial hypertension was demonstrated to determine a dysfunction in the whole LV systolic dynamic that can be detected even at very early stages by multiple ultrasound tools. These alterations are all highly prevalent in patients with HFpEF and have been demonstrated to have a strong prognostic role in this setting of patients [90,91].

## 5. Abnormal Renal Function and Microalbuminuria

The kidney is a target organ of hypertension. In fact, microalbuminuria (MAU), proteinuria, and CKD are considered the renal manifestations of HMOD and are associated with enhanced risk of major cardiovascular (CV) events and reduced life expectancy. In particular, CKD has been associated with a higher risk of developing HF [92]. For this reason, guidelines for the management of arterial hypertension highly recommend the identification of renal damage for the stratification of CV risk [1,93].

MAU is defined as the urinary albumin/creatinine ratio of 30–300 mg/g, and it is usually present in 2.2–11.8% of the general population [94] and in 26–58% of patients with arterial hypertension [95]. This large variability is due to the different clinical characteristics and ethnicity of individuals included in the studies that assessed the incidence/prevalence of MAU. For instance, considering the high prevalence of MAU in patients with type 2 diabetes (T2D) and the tight association between T2D and hypertension [96], patients with T2D and hypertension are expected to have a high incidence of MAU. One of the most controversial aspects of MAU is its role as a determinant of CV risk in patients with hypertension. In fact, a retrospective analysis of 141 patients with hypertension who were followed up for seven years documented a higher incidence of CV events and a greater impairment of renal function in those presenting MAU [97]. This finding was confirmed by a prospective study performed on 2085 subjects with hypertension or pre-hypertension, documenting that MAU increased the risk of ischemic heart disease (IHD) by fourfold [98]. However, these results were only partially confirmed by more recent studies. In fact, a prospective study showed in a cohort of 804 patients who were followed up for 3 years, that MAU was not associated with an increased risk of CV events in the entire study population but only in patients with previous IHD [99]. These data were consistent with the results of the INSIGHT (INvestigation of patients with ischemic Stroke In neuroloGic reHabiliTation) registry, which documented that MAU was associated with a higher risk of recurrent CV events in patients with acute stroke [100], suggesting the key role of MAU in the stratification of residual risk [101].

It should be underlined that MAU is associated with different mechanisms which are involved in the pathogenesis of CV events, such as mechanical stress, endothelial dysfunction, vascular inflammation, insulin resistance, hypercoagulability, etc. Thus, MAU, rather than being viewed merely as an initial manifestation of hypertensive nephropathy, should be considered as an expression of impaired control of CV homeostasis. Similarly, the development of CKD in hypertension, rather than being considered only as the consequence of hemodynamic overload, should be recognized as a complex and multifactorial process involving neuro-hormonal activation, inflammatory response, oxidative stress, etc. Interestingly, these pathogenic mechanisms also account for the development of other manifestations of HMOD, such as LVH, HFpEF, atherosclerosis, non-alcoholic fatty liver disease (NAFLD), cerebral lacunae, and retinopathy. Thus, it is not surprising that the different forms of HMOD often coexist. For instance, the combination of LVH + CKD has been found in 8–16% of patients with hypertension [3,102]. The clinical implication of this association is relevant since it can participate in the development of HFpEF. In this regard, it is noteworthy that MAU has been proposed as a biomarker of HFpEF [103]. In addition, MAU was documented to play a role in predicting the prognosis of patients with HFpEF [104].

In this scenario, the preservation as well as the restoration of normal renal function represent an important target of antihypertensive therapy. Several studies have tested the efficacy of different antihypertensive agents on renal targets, such as the time of doubling of serum creatinine value, the reduction of MAU or proteinuria, and the increase of glomerular filtration rate. These studies and the meta-analyses of randomized controlled trials have unequivocally documented the major efficacy of the inhibitors of the renin-angiotensin-aldosterone system in the prevention of renal damage, and, in general, of HMOD [105,106], indicating that these classes of drugs should always be the first choice in anti-hypertensive therapy.

## 6. Treatment Strategies for HFPEF

HFpEF represents a leading cause of morbidity and mortality worldwide. The global population is becoming older. In 2017, the world population aged 60 or older reached approximately 1 billion, and this figure is projected to triple by 2050. Based on this, it is anticipated that the prevalence of HF, and HFpEF in particular, would increase gradually over the next 20 years. Current guidelines do not recognize any specific treatment that reduces mortality and morbidity in patients with HFPEF. Thus, the most effective strategies would be to prevent its development starting with aggressive pharmacological treatment of arterial hypertension. The strategy to be pursued would act to prevent first development of structural and functional HMOD, taking into consideration that its regression is difficult to achieve. Recent validation of asymptomatic LVH as a possible risk factor for HFpEF has been proposed to bolster this conclusion [84]. Specifically, a retrospective cohort study revealed that patients with asymptomatic LVH developed HFpEF after a median follow-up of 8 years, which was associated with the worsening of LV diastolic function, whereas the transition to HFrEF was infrequent and characterized by only mild LV systolic dysfunction [84]. LVH, coronary microvascular dysfunction, subclinical systolic dysfunction, and kidney disease are the main pathophysiological issues related to the development and maintenance of HFpEF. Once HFpEF is diagnosed, treatment for hypertension is recognized as the main therapeutic goal with strict targets as outlined by recent American guidelines on HF [107].

In contrast to individuals with HFrEF, the pharmaceutical therapy of HFpEF remains difficult. Regarding RAAS inhibitors, none of the big RCTs undertaken in HFpEF have met their main outcomes, including the PEP-CHF (perindopril) [108], CHARM-Preserved (candesartan) [109], I-PRESERVE (irbesartan) [110], and TOPCAT (spironolactone) [111] trials. However, subgroup analysis from the PARAGON-HF study revealed a decrease in HF hospitalizations in individuals with an EF < 57%, and a pooled analysis of the PARADIGM-HF and PARAGON-HF trials revealed a decrease in CV mortality and HF hospitalization in those with an EF below the normal range.

Latest research has also recognized that sodium-glucose cotransporter 2 inhibitor (SGLT2-i) is an effective strategy in reducing combined end point for hospitalization and death, while sacubitril/valsartan are effective in reducing the hospitalization for heart failure [112,113].

As of now, 68 studies with active recruitment have been identified on https://clinicaltrials.gov/ accessed on 14th August 2022, with different therapeutic targets focusing on comorbidities, pulmonary hypertension, and specific molecular targets.

## 7. Conclusions

Patients with hypertension have a high risk for developing HFpEF. The presence of LVH is strongly associated with coronary microvascular dysfunction, which in turns leads to subclinical LV systolic impairment and development of HFpEF. Echocardiographic assessment is of paramount importance for early detection of any structural and/or functional HMOD.

## Figures and Tables

**Figure 1 jcm-11-05377-f001:**
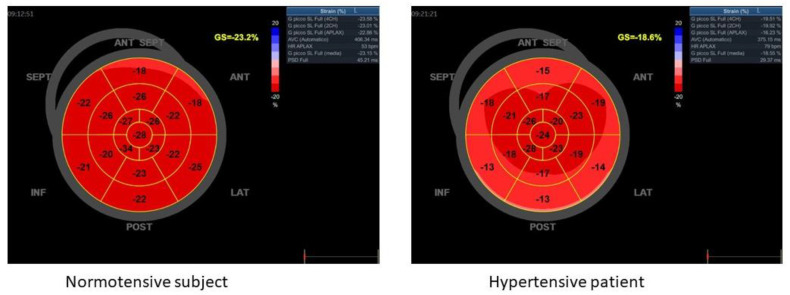
GLS in normotensive and hypertensive patient.

**Table 1 jcm-11-05377-t001:** Limitations of ejection fraction.

Problem	Inaccuracy-Related Conditions	Potential Solution
Geometry reliance	LBBB, significant abnormalities of wall motion, off-axis imaging	3D imaging, geometry-independent approaches
Load reliance	Extreme afterload values and mitral regurgitation	Pressure volume loops, pre-ejection markings
High vs. low HR	Heart block and tachycardias	None
Endocardial shortening marker	LVH	Mid-myocardial shortening
Insensitivity to slight modifications	Prognostic value close to EF 50%	Methods besides EF for evaluating subclinical dysfunction
Expertise	Common usage of EF	Quantification

**Table 2 jcm-11-05377-t002:** Structural, Functional, and Ultra-structural left ventricle differences between HFpEF and heart failure with reduced ejection fraction (HFrEF).

	HFpEF	HFrEF
LV Structure/Function		
End-diastolic volume	≃	↑
End-systolic volume	≃	↑
Wall thickness	↑	≃
Mass	↑	↑
Mass/volume ratio	↑	↓
Remodeling	Concentric	Eccentric
Ejection fraction	≃	↓
Stroke work	≃	↓
End-systolic elastance	≃	↓
End-diastolic stiffness	↑	↓
**LV Ultrastructure**		
Myocyte diameter	↑	≃
Myocyte length	≃	↑
Myocyte remodeling	Concentric	Eccentric
Fibrosis	Interstitial/reactive	Focal/replacement

Legend: “≃” not influenced, “↑” increased, “↓” reduced.

## Data Availability

Not applicable.

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
