# Peer review of "From Structural to Functional Hypertension Mediated Target Organ Damage—A Long Way to Heart Failure with Preserved Ejection Fraction"

_jcm, 2022, doi:10.3390/jcm11185377_

Round 1

Reviewer 1 Report

   The highlights of this review article are important for the diagnostic and therapeutic management of these patients (focus on the useful diagnostic tools for the LVH quantification and the detection of the subclinical systolic dysfunction). Please specify the impact of LVH on the left atrium remodelling (mechanical and electrical), as an "arrhythmogenic" factor, predisposing to atrial fibrillation that influence the prognosis.

Author Response

Thanks for your comment. We have added the following section as requested:

“LVH contribute to the remodelling of left atrium, acting a trigger for the development and maintenance of atrial fibrillation (AF)[1]. Patients with concentric LVH have more significant diastolic dysfunction leading to chronic elevation of filling pressures which in turns results in left atrial dilatation and development of AF[2,3]. Furthermore different patterns of LVH have been demonstrated to be important determinants of cardiovascular outcomes in patients with AF contributing to the development of adverse cardiovascular outcome including overt HF[4].

Reviewer 2 Report

Extremely important topic! An excellent review article, contemporary and comprehensively presented the importance of hypertension as one of the leading risk factors for the development of cardiovascular diseases!

I suggest that the authors go over the entire article again, to correct a few typos, spacing between words, and similar errors. I believe it has already been done.

Author Response

Thanks for your comment

Reviewer 3 Report

I read with interest the article entitled «From Structural to Functional Hypertension Mediated Target Organ Damage. A Long Way to Heart Failure with Preserved Ejection Fraction» which concerns an important and clinically interesting issue. The manuscript contains valuable comments and information. However, I have several comments regarding this manuscript

Major:

1) Manuscript should undergo significant English revisions

2) I strongly recommend to add a separate section on LV diastolic dysfunction in LVH, that is a major structural and functional contributor of increased filling pressures and the development of HFpEF

3) There are no figures and tables in the article, which impoverishes the perception of the article. It is reasonable to insert figures with examples of GLS, MEEi, 3D-strain, etc.; tables on comparison of different methods of LV systolic function estimation, coronary reserve, pharmacotherapies for treating LVH/HFpEF etc.

Minor:

1) Authors present many facts, but poorly present possible causes and mechanisms. For example, after the statement "Circumferential and radial strain deformations … resulted to be impaired in more advanced stages of hypertension, being more pronounced after development of LVH" it would be important to offer a hypothesis of such a fact.

2) In Systolic dysfunction section, it is reasonable to consider in more details the mechanisms of systolic dysfunction development in LVH (apoptosis, ischemia, "burned-out" left ventricle etc)

3) In the same section, it is reasonable to consider the clinical data on the natural history of LV systolic dysfunction in hypertensive heart disease (e.g. Am. J. Cardiol. 2004, 93, 234-237; J. Am. Coll. Cardiol. 2004, 43, 2207-2215; Am. J. Cardiol. 2011, 108, 997-1001; J. Clin. Med. 2022, 11(13), 3885).

4) In Coronary Microvascular Dysfunction section, it is appropriate to include information on the PROMIS-HFpEF trial (Eur. Heart J. 2018, 39, 3439-3450), demonstrated a high prevalence of coronary microvascular dysfunction in HFpEF in the absence of macrovascular coronary artery disease

5) In the same section, it is advisable to present in more detail the experimental and clinical data on the therapy of microvascular dysfunction in LVH/HFpEF (ARNI, agents targeting NO-cGMP-PKG axis, IL1 inhibitors, SGLT2 inhibitors, endothelin receptor antagonists etc).

6) Treatment strategies for HFPEF section is written too briefly. It is advisable to expand this section with data on the regression of HF with antihypertensive drugs, studies on the prevention of transition from asymptomatic hypertensive heart disease to HF (e.g. J. Clin. Med. 2022, 11(13), 3885), the treatment of HFpEF associated with LVH, etc.

7) Line 72 – In the H2FPEF algorithm for assessing the likelihood of a presence of HFpEF, indeed, there is no LVH; but in recent recommendations for the diagnosis of HFpEF (ESC-2021, HFA-PEFF, 2019), LVH is an important contributor to the diagnosis

8) Line 80 – reference is needed

9) Line 93 – reference is needed

10) Line 96 – reference is needed

11) Line 98 – the repetition of it

12) Line 125 – comma instead of dot

13) Lines 144-148 – the repetition of "indexed for ... mass"

14) Lines 166-171 – the sentence is too long and hard to understand, it is advisable to split it into two sentences

Author Response

I read with interest the article entitled «From Structural to Functional Hypertension Mediated Target Organ Damage. A Long Way to Heart Failure with Preserved Ejection Fraction» which concerns an important and clinically interesting issue. The manuscript contains valuable comments and information. However, I have several comments regarding this manuscript

Major:

  • Manuscript should undergo significant English revisions

This has been done.

I strongly recommend to add a separate section on LV diastolic dysfunction in LVH, that is a major structural and functional contributor of increased filling pressures and the development of HFpEF Thanks for your comment. The following section has been added:

“The impact of LVH on diastolic dysfunction has demonstrated by several studies focusing of general population and hypertensive patients and is associated with worsened cardiovascular prognosis[5,6]. Myocardial hypertrophy leads to an increase of the myocardial mass/volume ratio, and the degree of hypertrophy is the main determinant of chamber stiffness. The main, if not unique, determinant of myocardial diastolic tissue distensibility is the structure and concentration of the collagen. LV collagen concentration is elevated due to reactive fibrosis. Studies using MRI have demonstrated that development of fibrosis proceed the overt HF and might be considered as important contributor to its development[7]

  • There are no figures and tables in the article, which impoverishes the perception of the article. It is reasonable to insert figures with examples of GLS, MEEi, 3D-strain, etc.; tables on comparison of different methods of LV systolic function estimation, coronary reserve, pharmacotherapies for treating LVH/HFpEF etc.

 We thank the reviewer for the valuable suggestion. Figure 1, tables 1 and 2 have been added to the manuscript.

Minor:

  • Authors present many facts, but poorly present possible causes and mechanisms. For example, after the statement "Circumferential and radial strain deformations … resulted to be impaired in more advanced stages of hypertension, being more pronounced after development of LVH"it would be important to offer a hypothesis of such a fact.

We thank the reviewer for the criticism. In the new draft of the manuscript, we added that “This phenomenon seems related to the progressive accumulation of pathologic fibrosis and disarray of connective tissue matrix of interstitial collagen fibers in LVH, impacting on LV circumferential contraction and disturbing radial thickening”.

  • In Systolic dysfunction section, it is reasonable to consider in more details the mechanisms of systolic dysfunction development in LVH (apoptosis, ischemia, "burned-out" left ventricle etc).

We thank the reviewer for the valuable suggestion. In Systolic dysfunction section, we further discuss this point and stated that “LVH induced by arterial hypertension is accompanied by vascular and interstitial remodelling and fibrosis leading to progressive augmented microvascular resistance and arterial stiffness, endothelial disfunction and impaired capillary density, also affecting the coronary district. All those abnormalities could contribute to trigger myocardial remodelling and apoptosis, myocardial ischemia and progressively determine LV dilation and drop of LV systolic performance”.

  • In the same section, it is reasonable to consider the clinical data on the natural history of LV systolic dysfunction in hypertensive heart disease (e.g.  J. Cardiol.2004, 93, 234-237; J. Am. Coll. Cardiol. 2004, 43, 2207-2215; Am. J. Cardiol. 2011, 108, 997-1001; J. Clin. Med. 2022, 11(13), 3885).

We thank the reviewer for the kind suggestion. In the new draft of the manuscript we added “Actually, the natural history of hypertensive heart disease is related to the lost balance of what started as a compensatory mechanism to pressure overload and evolved towards progressive stages including abnormal LVH, diastolic impairment, HFpEF, LV dilation and heart failure with reduced ejection fraction” and cited the suggested references.

  • In Coronary Microvascular Dysfunction section, it is appropriate to include information on the PROMIS-HFpEF trial (Eur. Heart J. 2018, 39, 3439-3450), demonstrated a high prevalence of coronary microvascular dysfunction in HFpEF in the absence of macrovascular coronary artery disease.

Thanks for your comment. The suggested citation has been added

  • In the same section, it is advisable to present in more detail the experimental and clinical data on the therapy of microvascular dysfunction in LVH/HFpEF (ARNI, agents targeting NO-cGMP-PKG axis, IL1 inhibitors, SGLT2 inhibitors, endothelin receptor antagonists etc).

Thanks for your comment. The following sentence has been added to highlight your point:

“Recent studies have demonstrated the potential role of specific pharmacological therapy for microvascular dysfunction in HFpEF. In particular data from animal studies have demonstrated that agents targeting NO-activated guanylyl cyclase (GC) might have beneficial effect on microvascular dysfunction in HFpEF [8]. Recently vericiguat, a once-daily stimulator of sGC, was tested in population of patients with preserved systolic function. Unfortunately the primary endpoint, defined as the change in NT-proBNP level, was negative, but there was an evidence of improved physical capacity and quality of life [9]. Promising therapy with SGLT2 inhibitors have demonstrated to improve microvascular dysfunction in animal models acting through the nitric oxide pathway  [10]

  • Treatment strategies for HFPEF section is written too briefly. It is advisable to expand this section with data on the regression of HF with antihypertensive drugs, studies on the prevention of transition from asymptomatic hypertensive heart disease to HF (e.g. J. Clin. Med. 2022, 11(13), 3885), the treatment of HFpEF associated with LVH, etc.

 We thank the reviewer for the valuable suggestion. We have expanded the HFpEF section as requested.

  • Line 72 – In the H2FPEF algorithm for assessing the likelihood of a presence of HFpEF, indeed, there is no LVH; but in recent recommendations for the diagnosis of HFpEF (ESC-2021, HFA-PEFF, 2019), LVH is an important contributor to the diagnosis

Thanks for your comment. The following sentence has been added to highlight your point.

“The more recent guideline of HF by ESC highlights the importance of LV mass as a main morphological criteria for the diagnosis of HFpEF” (8).

8) Line 80 – reference is needed. The reference has been added [11]

9) Line 93 – reference is needed The reference has been added  [12]

10) Line 96 – reference is needed The reference has been added [12]

11) Line 98 – the repetition of it Corrected

12) Line 125 – comma instead of dot Corrected

13) Lines 144-148 – the repetition of "indexed for ... mass" Corrected

14) Lines 166-171 – the sentence is too long and hard to understand, it is advisable to split it into two sentences Done.

Round 2

Reviewer 3 Report

Dear authors, thank you for thorough revision of the manuscript; in its present form the article can be accepted